# A community health worker-led program to improve access to gestational diabetes screening in urban slums of Pune, India: Results from a mixed methods study

Puja Chebrolu[1]*, Andrea Chalem[2], Matthew Ponticiello[3], Kathryn Broderick[1], Arthi Vaidyanathan[1], Rachel Lorenc[1], Vaishali Kulkarni[4], Ashlesha Onawale[4], Jyoti S. Mathad[1☯], Radhika Sundararajan[1☯]

1 Center for Global Health, Weill Cornell Medicine, New York, New York, United States of America, 2 Department of Epidemiology, Gillings School of Global Public Health, University of North Carolina, Chapel Hill, North Carolina, United States of America, 3 Department of Medicine, Yale University School of Medicine, New Haven, Connecticut, United States of America, 4 Deep Griha Society, Pune, Maharashtra, India

☯ These authors contributed equally to this work.
* puc9005@med.cornell.edu

**Data Availability Statement:** The data for this study cannot be publicly shared as they contain

## Abstract

The World Health Organization recommends all pregnant women receive screening for gestational diabetes (GDM) with a fasting oral glucose tolerance test (OGTT). However, very few women receive recommended screening in resource-limited countries like India. We implemented a community health worker (CHW)-delivered program to evaluate if home-based, CHW-delivered OGTT would increase GDM screening in a low-resource setting. We conducted a mixed methods study in two urban slum communities in Pune, India. CHWs were trained to deliver home-based, point-of-care fasting OGTT to women in their third trimester of pregnancy. The primary outcome was uptake of CHW-delivered OGTT. Secondary outcomes included GDM prevalence and linkage to GDM care. Individual interviews were conducted with purposively sampled pregnant women, CHWs, and local clinicians to assess barriers and facilitators of this approach. From October 2021-June 2022, 248 eligible pregnant women were identified. Of these, 223 (90%) accepted CHW-delivered OGTT and 31 (14%) were diagnosed with GDM. Thirty (97%) women diagnosed with GDM subsequently sought GDM care; only 10 (33%) received lifestyle counseling or pharmacologic therapy. Qualitative interviews indicated that CHW-delivered testing was considered highly acceptable as home-based testing saved time and was more convenient than clinic-based testing. Inconsistent clinical management of GDM was attributed to providers' lack of time to deliver counseling, and perceptions that low-income populations are not at risk for GDM. Convenience and trust in a CHW-delivered GDM screening program resulted in high access to gold-standard OGTT screening and identification of a high GDM prevalence among pregnant women in two urban slum communities. Appropriate linkage to care was limited by clinician time constraints and misperceptions of GDM risk. CHW-delivered GDM screening and counseling may improve health education and

information that can potentially identify some of the participants. However, de-identified participant data will be made available with a signed data access agreement upon request to Megan Urry (meu7003@med.cornell.edu).

**Funding:** This work was financially supported by the Weill Cornell Department of Medicine Primary Care Innovation Grant awarded to JSM and RS. This work was also financially supported by the Weill Cornell Department of Medicine Fund for the Future Grant awarded to PC. This work was also financially supported by the National Institutes of Health/Fogarty International Center (D43 TW009337) awarded to PC. This work was also financially supported by the Bonnie Johnson Sacerdote Clinical Scholarship in Women's Health awarded to JM. This work was also financially supported by the National Institutes of Health/ National Institutes of Allergy and Infectious Disease (K23 AI129854) awarded to JM. This work was also financially supported by the National Institutes of Health/National Center for Advancing Translational Sciences in the form of a grant (UL1 TR 002384) awarded to Weill Cornell Medicine. The funders had no role in study design, data collection and analysis, decision to publish, or preparation of the manuscript.

**Competing interests:** The authors have declared that no competing interests exist.

access to preventive healthcare, offloading busy public clinics in high-need, low-resource settings.

## Introduction

Gestational diabetes mellitus (GDM)–glucose intolerance that develops during pregnancy–affects 10% of pregnancies worldwide [1]. Women with GDM are at high risk of pregnancy complications such as pre-eclampsia, C-section, and birth asphyxia. More than 35% of Indian women with GDM will develop type 2 diabetes (T2DM) within 5 years of delivery, leading to a three-fold higher mortality compared with the general population of India [2, 3].

Early diagnosis and management of GDM is essential to decrease perinatal complications and prevent the long-term sequelae of GDM on maternal and child health. For this reason, the World Health Organization (WHO) recommends GDM screening in the $2^{nd}$ or $3^{rd}$ trimester of pregnancy [4]. The fasting oral glucose tolerance test (OGTT), which is generally administered in antenatal clinics, is preferred for screening. Unfortunately, most pregnant women do not complete an OGTT because of logistical and financial challenges, such as traveling by public transportation, losing wages, paying for travel, waiting in long queues while pregnant and fasting, and having other caretaking responsibilities [5, 6]. A study in Tanzania found uptake of OGTT to be only 3.4% at a major tertiary care hospital [7]. Even in high-income countries, OGTT uptake ranges from only 18% to 37% [8–11].

Task-shifting from clinical personnel to laypersons has been shown to improve access to evidence-based health services in low- and middle-income countries [12, 13]. Community health workers (CHW) have successfully improved screening for HIV, hypertension, and tuberculosis among nonpregnant populations in low-resource settings [14–18]. Task shifting of preventative interventions are also recommended by the WHO in pregnancy to improve the utilization and quality of antenatal care [19]. Therefore, we hypothesized that home-based GDM screening in two urban slum communities of Pune, India would improve OGTT uptake by decreasing the logistical barriers to administering the fasting OGTT. Secondary objectives included determining GDM prevalence, linkage to GDM care, and conducting a qualitative evaluation of the barriers and facilitators of home-based OGTT.

## Methods

### Study setting and design

This study was conducted in two slum communities of Pune, India: Tadiwala Road (TR) and Yerawada Block (YB) (Fig 1). Geographic delineations of these communities were based on the most recent Indian Census [20]. The study was conducted from October 2021 to June 2022 in collaboration with the Deep Griha Society, a local non-governmental organization with social service programs in Pune's slum communities.

### Sampling and recruitment

CHWs went from door-to-door within the study areas to recruit pregnant participants and screen them for eligibility. Each community of approximately 80,000–100,000 people was divided into 5 geographic sub-areas [21]. Within each sub-area, one CHW systematically screened each household for eligible pregnant women. The total number of pregnant women in each area was estimated using birth rate data [22].

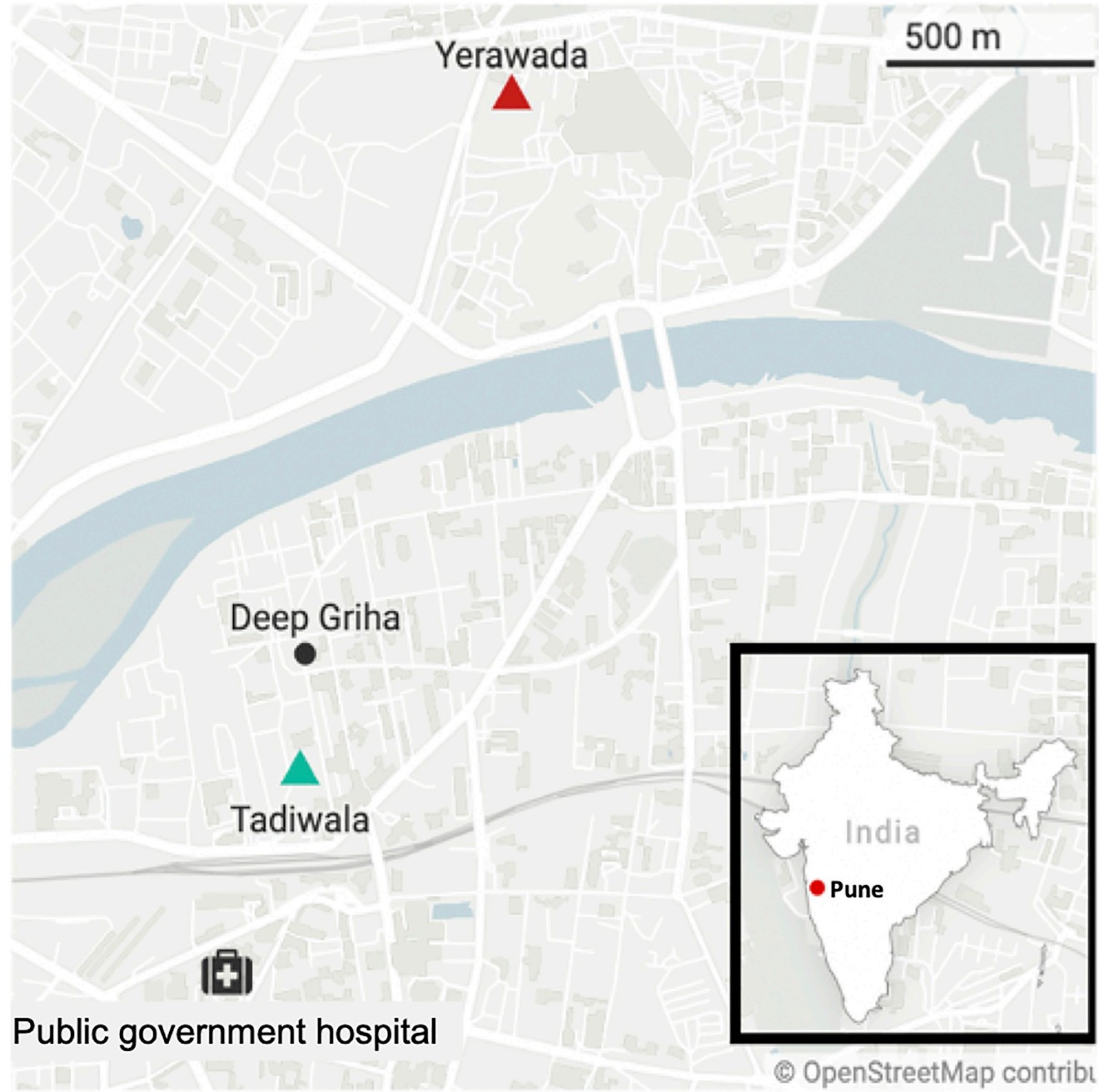

**Fig 1. Map of study setting.** Recruitment occurred in Yerawada Block and Tadiwala Road communities. The government hospital is where many of the pregnant women living in these communities receive prenatal care.

Inclusion criteria included: 1) residence in TR or YB; 2) age ≥18 years; 3) gestational age ≥24 weeks; 4) no previous fasting OGTT for GDM during the current pregnancy; 5) and no prior history of diabetes. Gestational age was assessed using a pregnancy wheel and last menstrual period. If the participant was unable to remember last menstrual period, gestational age was calculated using available ultrasound reports. Following enrollment, CHWs instructed participants to remain fasting overnight. Fasting was defined as no oral intake other than

 

water and medications for the 8 hours prior to the visit. The CHW then returned the following morning to administer the OGTT.

## Data collection

Demographic information, health history, vital signs, and anthropometric data were collected at study enrollment. Data was collected using the REDCap mobile app and synced to the secure online REDCap server at the end of each day of data collection. The REDCap server belonged to Weill Cornell Medicine and access was provided to study investigators only. CHWs did not have access to the data after syncing.

## CHW training

A one-day training session was conducted. CHWs received training from a nurse from the local government hospital on calculating gestational age, measuring vital signs and anthropometrics. CHWs also received training on gold standard practices for administering the point-of-care fasting OGTT delivered by an obstetrician from the local government hospital. We produced an educational flyer summarizing guidelines to be provided to participants with GDM. CHWs were trained on GDM counselling using this flyer.

## OGTT administration and interpretation

CHWs administered the OGTT in the morning at a convenient time scheduled with study participants. On arrival, the CHW measured fasting glucose via capillary blood sample using the Contour Plus One glucose monitor. CHWs then dissolved a powdered 75g glucose packet in 300mL of bottled water. The participant was instructed to drink this within 10 minutes. During a two-hour waiting period, participants watched educational videos about GDM and antenatal care. After two hours, glucose was rechecked. GDM was defined as fasting glucose >91 mg/dl and/or 2-hour post-OGTT glucose >152 mg/dl per International Association of Diabetes in Pregnancy Study Group (IADPSG) criteria [23]. Pregnant women were provided the results of their OGTT and relevant counseling upon completion of the test.

## Linkage to GDM care and follow-up

Women who met criteria for GDM or had abnormal vital signs were provided a referral letter to take to the local government antenatal clinic. The referral letter contained details of their test results. The Sassoon General Hospital antenatal clinic, is government-run and provides free care. It is within 5 kms of both sites. After two weeks, CHWs followed up in person or via phone call with women diagnosed with GDM to determine if they had sought GDM care. If a participant had attended a clinic for care, they were asked about their experiences at the clinic, including any GDM counseling or pharmacologic therapy. Women who met criteria for other abnormalities in vital signs were referred to care through a tiered process based on the severity of the abnormality.

## Qualitative interview

The objectives of the interviews were to define barriers and facilitators to CHW-led screening.

**Sampling strategy.** A subset of 30 pregnant women, all five CHWs, and 18 clinicians (obstetrics and general medicine clinicians practicing in Pune) were invited to participate in a single semi-structured qualitative interview. Participants were selected through purposive sampling representing the primary and secondary study outcomes as well as recruitment sites. Inclusion and exclusion criteria were as described above. Clinicians were invited from facilities

where pregnant women diagnosed with GDM reported seeking care. Inclusion criteria included age >18 years and ability to provide informed consent. All individuals approached to participate in a qualitative interview agreed to participate.

**Data collection procedures.**  For all interviewees, a native Marathi (the local language)-speaking interviewer trained in social science and qualitative methods asked about barriers and facilitators to CHW-delivered GDM screening using a pre-prepared interview guide [S1 and S2 Texts]. Specifically, pregnant women were asked to discuss experiences interacting with and receiving screening from CHWs, and linkage to GDM care if pertinent. CHWs were asked about their experience providing GDM screening and counseling. Clinicians were asked about their perceptions of GDM prevalence, diagnosis, and treatment. Interviews were audio recorded, transcribed, and translated from Marathi into English by a professional translation service.

## Sample size and power

A priori, we estimated that enrolling 200 pregnant women would provide at least 80% power and 95% confidence to detect an improvement in screening rate from 2% (historical data) to 50% with this intervention. Even if the CHW screening rate was only 20%, we would still have >80% power to detect a significant difference in screening. Post-hoc, the sample size needed to detect the screening uptake rate we found in this study was 139.

For qualitative data, sample size was guided by the concept of data saturation, when interview content no longer provides novel concepts [24]. We estimated data saturation after 35 interviews were completed.

## Data analysis

**Quantitative data.**  The primary study outcome was uptake of CHW-delivered OGTT. This was defined as the proportion of eligible women accepting and completing a fasting OGTT at home, delivered by the CHW. Secondary outcomes included prevalence of GDM and linkage to care. Linkage to care was defined as having attended at least one clinic visit for GDM. Socioeconomic status was categorized according to the Kuppuswamy scale [25]. Descriptive statistics were performed on baseline characteristics of participants who accepted the intervention. Statistical analysis was performed using Stata version 15.1 (College Station, TX).

**Qualitative data.**  Qualitative data were analyzed using an inductive, content-analysis approach [26–28]. All transcripts were reviewed to develop a coding scheme relevant to screening uptake, and to illuminate barriers and facilitators to the CHW-delivered strategy. Codes were independently developed by four authors (KB, MP, AC, RL) in vivo through repeated engagement with the dataset; disagreements or discrepancies in codes were resolved through discussion between these authors. Using a framework approach, coded data was organized by topic, and entered into an analytical matrix by KB. KB and RS reviewed the matrix to identify larger concepts that could speak to the acceptability and scalability of this approach.

## Ethics approvals

This study was approved by the Weill Cornell Medicine Institutional Review Board and Sahara Aalhad, an Institutional Review Board in Pune, India. Written informed consent was obtained from all study participants in Marathi. All interviews were conducted in secure, private locations to maintain participant confidentiality.

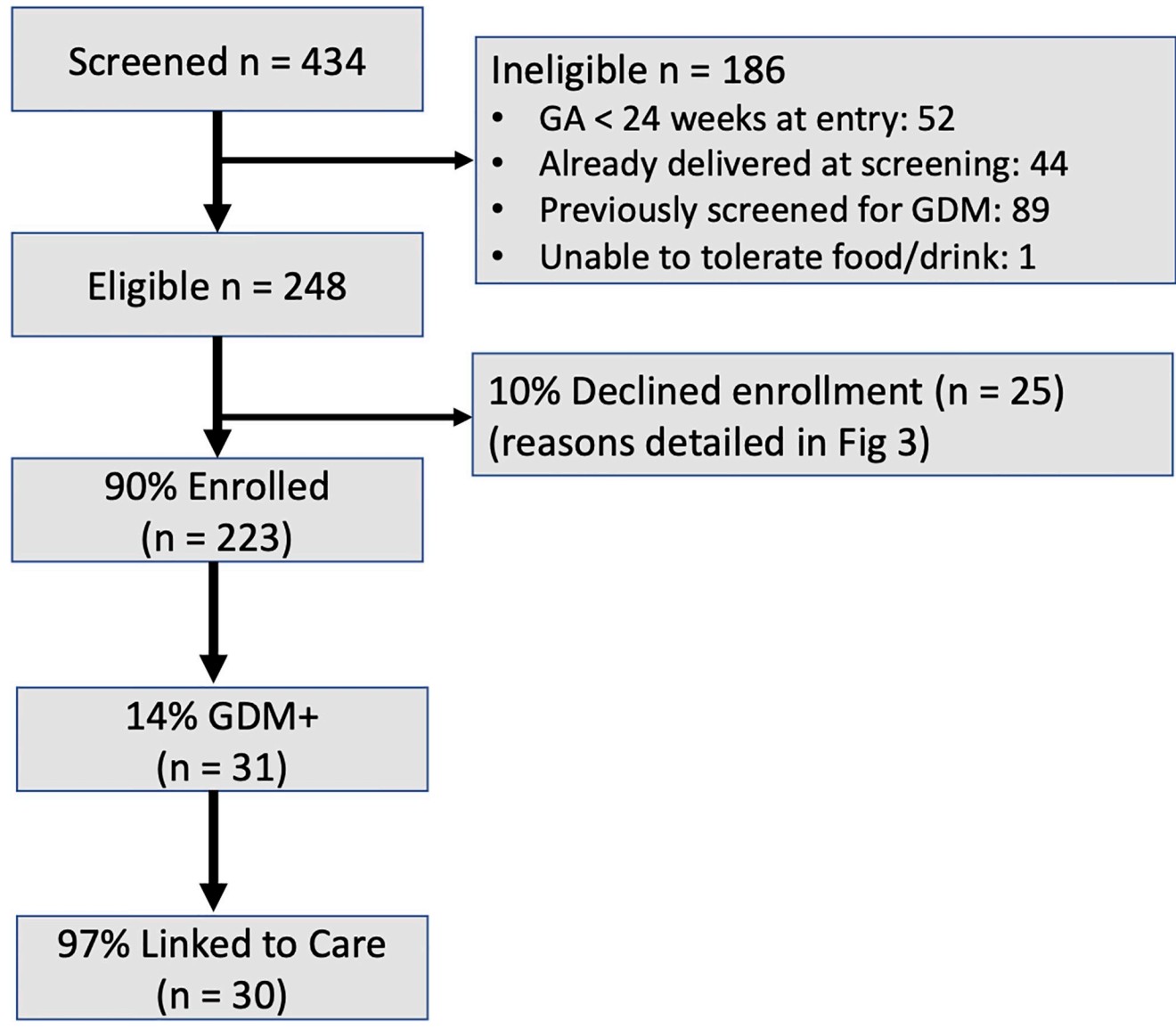

**Fig 2. Study flowchart.**

## Results

### Screening uptake and GDM prevalence

Overall, 434 pregnant women were approached for enrollment during the study period (Fig 2). Of 248 eligible women, 223 (90%) underwent CHW-delivered fasting OGTT. Table 1 shows the characteristics of the 223 pregnant women who accepted the CHW-delivered OGTT. The median age was 24 years (IQR 22–27) and median gestational age was 26 weeks (IQR 24–31). Almost half of the participants had received at least a high school diploma (48%) and most participants were of upper-lower or lower-middle socioeconomic status (41% and 53%, respectively). There was a family history of diabetes in 27 participants (12%). At enrollment, 28 (13%) participants had a blood pressure >130/80 mmHg. The median BMI of the cohort was 23.8 kg/m$^2$ (21.4–27.4). In total, 31 (14%) participants were diagnosed with GDM.

**Table 1. Characteristics of participants with and without GDM.**

| Characteristic | GDM (n = 31) N (%) | Non-GDM (n = 192) N (%) |
|---|---|---|
| Age, median (IQR) | 23 (21–27) | 24 (22–27) |
| Gestational age, weeks, median (IQR) | 25 (24–29) | 26 (24–31.5) |
| Known family history of diabetes | 6 (19.35) | 21 (10.94) |
| Hypertension (>130/80 mmHg) | 7 (22.58) | 21 (10.94) |
| BMI, kg/m$^2$, median (IQR) | 25.5 (21.7–30.3) | 23.5 (20.9–27.2) |
| Mid-upper arm circumference, cm, (IQR) | 27 (25–30) | 26 (24–28) |
| Waist circumference, cm, (IQR) | 97 (90–110) | 96 (90–103) |
| Kuppuswamy socioeconomic status | | |
| *Upper-middle income* | 1 (3.33) | 13 (7.14) |
| *Lower-middle income* | 19 (63.33) | 67 (36.81) |
| *Upper-lower income* | 10 (33.33) | 102 (56.04) |

Reasons for inability to complete OGTT were as follows: patient or family preference for clinic-based testing (n = 9, 39.1%), completed a fasting OGTT during 1st trimester for T2DM screening (n = 8, 34.8%), desire to discuss OGTT with a doctor first (n = 4, 17.4%), fear of medical testing due to prior miscarriage (n = 1, 4.3%), unable to understand the procedure (n = 1, 4.3%), and not having time (n = 1, 4.3%) (Fig 3).

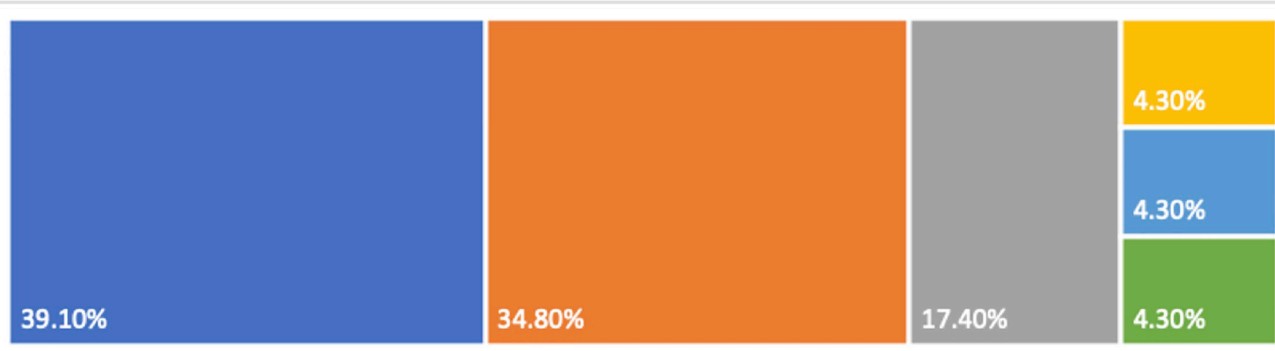

**Fig 3. Reasons provided among eligible participants who declined OGTT.**

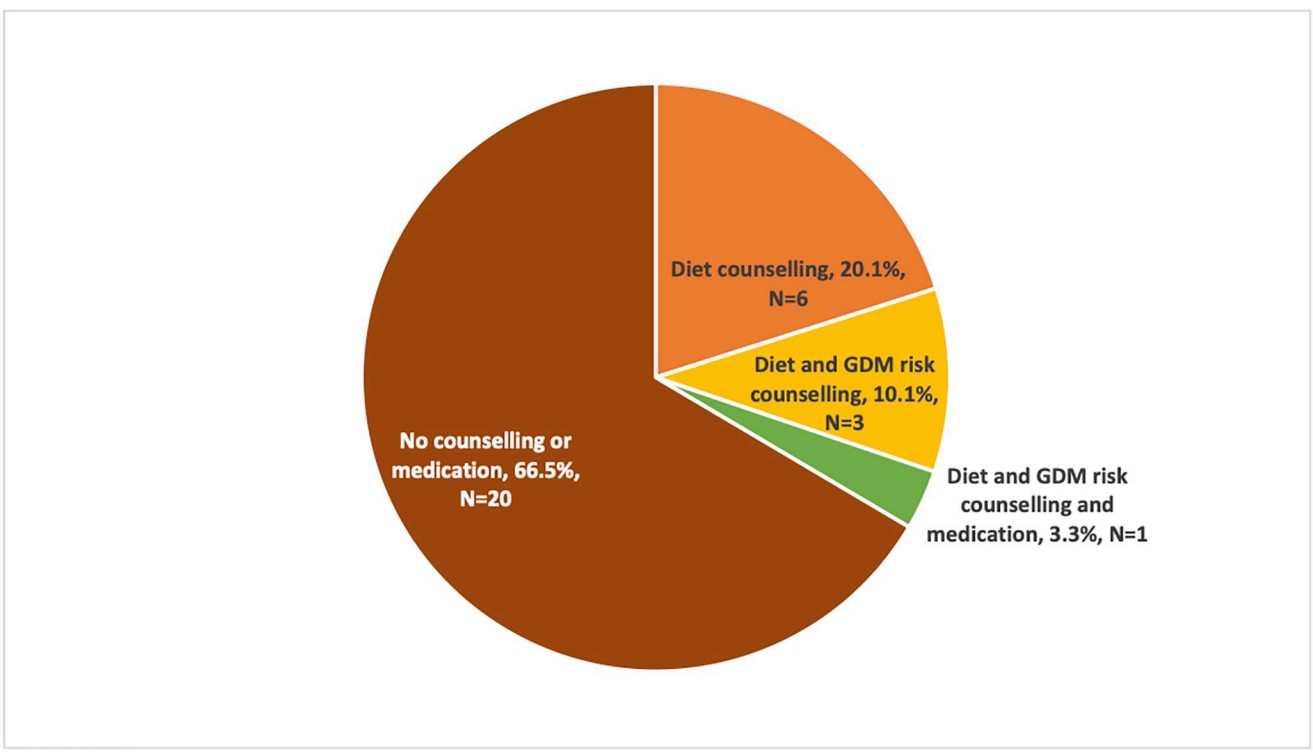

**Fig 4. Self-reported GDM management at clinic visit (n = 30).**

### Linkage to care

Of participants who were diagnosed with GDM, 30 of 31 (97%) women attended a clinic visit to discuss OGTT results. At the clinical visit, 20% (n = 6) received dietary counseling on GDM, 10% (n = 3) received diet and GDM risk reduction education, and 3% (n = 1) received diet counselling, risk reduction counseling, and medication for GDM. The majority (67%, n = 20) reported no counseling or medication changes related to their GDM diagnosis (Fig 4).

### Qualitative results

Our qualitative data provide explanation and context for the quantitative findings described above (Table 2). There were two primary factors driving high acceptability of the CHW-delivered among pregnant women: 1) the home-based OGTT overcame significant logistical barriers to clinic-based testing; and 2) pregnant women had a high level of trust in CHWs. Most women diagnosed with GDM as part of this study sought clinical care, reflecting effective CHW-delivered counseling on GDM as a health priority. We also noted that two factors explain inconsistent management of GDM in clinics: 1) perceived lack of time available to provide clinical counseling; and 2) the perception that low-income women are not at risk for GDM.

### CHW-delivered GDM screening was highly acceptable

Participants explained that home-based testing overcame barriers to reaching the antenatal care clinic, including lack of transportation. Pregnant women also noted that completing an OGTT in a clinical setting is time consuming, often requiring hours at the clinic. This time

**Table 2. Summary of trial results and supporting qualitative data from exit interviews.**

| Quantitative result | Qualitative theme | Representative quotes |
|---|---|---|
| GDM screening uptake was 90% via CHW-delivered OGTT | CHW testing mitigates logistical barriers to clinic-based care | *Because of the pregnancy, some ladies can't manage to travel to the clinic*—Pregnant woman, 24 years old, diagnosed with GDM |
| | CHW were perceived as trustworthy health advisors and counselors | *The things I didn't know about until now, I got to know about them —like [blood] sugar—from the* [29]*. . . And about diet, types of exercises, they explained about it properly"*—Pregnant woman, 28 years old, diagnosed with GDM |
| | | *"I said [to myself], she must be telling it to me for my own good"*—Pregnant woman, 28 years old, diagnosed with GDM |
| 97% of participants diagnosed with GDM sought follow up care | Participants were motivated by concern for the health of the fetus | *We are doing this [GDM screening] for you and your baby. We convinced [the participants] about this.*–CHW, 36 years old |
| Pregnant women with GDM were managed inconsistently at the clinics | Barriers to adequate clinical counseling include perceived lack of time, belief that individual counseling is ineffective, and perception that low-income populations have low GDM risk. | *A specialist would always be better [than me] because they can counsel properly. They can do more. . .. We can at the max give only 30 minutes [to our patients].*—General medicine physician, female, 1 year experience |
| | | *For overweight mothers . . . we do fasting OGTT and hemoglobin A1c [tests] for such patients.*—OB/Gyn physician, female, 22 years of experience |

commitment puts additional strain on pregnant women who have competing priorities such as childcare or household responsibilities. Participants found that the CHW-delivered screening minimized the inconvenience of receiving an OGTT in the typical clinical setting.

*The whole day goes by at the clinic for the OGTT. They used to tell me, the women from the hospital, 'get it done outside as far as possible, don't do it here' . . . I felt that it was really good that my time was saved.*

–pregnant woman, 23 years old, diagnosed with GDM

*One thing is it saved our time. . . As* [30] *come to our home, we don't have to do that much running around. . . [At the clinic], they ask you to go this number ward. You have to do running around for sure!. . . Then they don't allow relatives inside. You yourself have to go here and there."*

- pregnant woman, age 19, no GDM

As part of this study, pregnant women who accepted the OGTT had two or three in-person interactions with CHWs. CHWs were described as familiar, like "family". This sense of intimacy could be attributed to the fact that CHWs come from the same communities as the study participants.

*She was also nice to talk to . . . like a family member.*

- pregnant woman, 30 years old, no GDM

*Every time she enquired after me, 'How are you? Are you ok? Did you have a proper breakfast? Take care of yourself on time'. . . So I really did feel that she asks more about me than anyone in my family does.*

- pregnant woman, 33 years old, diagnosed with GDM

These trusting relationships facilitated effective health education and counseling on GDM.

*They say 'nobody explains it to us the way that you have explained it to us'*

*- CHW, 47 years old*

Deep Griha's long-standing community involvement also facilitated trust between CHWs and pregnant women.

*Many people know about Deep Griha. It's already familiar, and we know how they treat people over there. So that's why family members also didn't react negatively [to the CHW-delivered screening]. They said 'it's ok, if it is done by Deep Griha, then it will be done properly'.*

*- Pregnant woman, 20 years old, diagnosed with GDM*

*When you say the name Deep Griha you get a lot of support [from people living in these communities.].*

*- CHW, 36 years old*

## High rates of seeking GDM care following diagnosis

Before speaking with the CHW, pregnant women reported that they did not know about the risks or prevalence of GDM in their communities. Many said that screening had not been offered in the antenatal care clinic.

*The hospital people haven't even asked me to do the sugar test.*

*- pregnant woman, 24 years old, diagnosed with GDM*

Participants diagnosed with GDM were highly motivated to engage in clinical care, citing concern for the developing fetus as a motivator to seek GDM care.

*For the baby and the mother, [treatment] needs to be done. For the baby's good.*

*- pregnant woman, 31 years old, diagnosed with GDM*

## Inconsistent clinical management of GDM

Participants with GDM reported varied interactions with health care when they sought clinical care. Physicians in public clinic settings reported that the large number of daily clinical encounters made individual health counseling difficult.

*"Especially in the public hospital . . . where we have 100 patients and more, so it is not possible to do one-on-one counseling. We try and reserve counseling for specific people when they come to us"*

*- Physician, female, 22 years' experience*

Additionally, clinicians believed that counseling pregnant women on lifestyle changes would be ineffective without including the patients' families.

*"[Counseling is] not only for the patient. The family needs to be made aware of these things because the woman here is most dependent upon the family.*

*- Physician, female 30 years' experience*

While many clinicians believed that GDM incidence was rising in India, this change was attributed to an increase in maternal age, rates of obesity, sedentary lifestyles and poor diet. Therefore, many physicians believed that GDM was more of a concern in wealthier populations than in slum communities.

*"Less educated people get married at a very early age, so gestational diabetes is not that common. But those belonging from very well-educated families . . .. those girls are landing up with lot of gestational diabetes"*

*- Physician, female, 30 years' experience*

As such, screening for GDM was not routinely considered as standard antenatal healthcare for pregnant women who were not obese or of advanced maternal age.

## Discussion

We found that the CHW-delivered home GDM screening program was highly accepted among pregnant women in two low-income communities in Pune, India. This is important because we discovered that women in these communities have a higher prevalence of GDM than the national average in India (8.9%), suggesting that they are at especially high-risk for poor outcomes [31]. Nearly all women who were diagnosed with GDM visited a doctor for follow-up care with inconsistent subsequent clinical management. Taken together, our results indicate that CHWs can be trained to effectively provide gold standard GDM screening in the community, but additional work is needed to improve management of GDM.

Uptake of OGTT screening was high (90%) in the CHW-delivered program. A similarly high uptake of CHW-based care delivery has been seen in other studies for pregnant and non-pregnant populations [32, 33]. A study in Tamil Nadu, for example, found that a home-based cervical and oral cancer screening program was accepted by >90% of people approached [30]. In another study of >150,000 pregnant women in India, women contacted by CHWs had 60% greater utilization of antenatal care compared to women who were not contacted by CHWs [34]. Our qualitative data suggest that the high uptake of CHW-delivered OGTT was primarily due to mitigation of the logistical barriers to OGTT screening and the trust that pregnant women had in the CHWs. Therefore, our data suggest that, with appropriate training, even invasive procedures such as the OGTT may be integrated into existing CHW-delivered non-communicable disease and maternal health programs.

In India, government-employed CHWs (called ASHA workers) have been integral in improving the uptake of antenatal care among pregnant women. ASHA workers are women (aged 25–45) without a healthcare background who are trained to provide basic reproductive health and child health services to the communities in which they reside [35]. According to current Ministry of Health guidelines, ASHAs are tasked with coordinating GDM screening visits, accompanying women to clinic to get GDM screening done, explaining the OGTT process in detail, and even counseling women on their results. While Ministry of Health guidelines do not specifically prohibit ASHA workers from conducting blood testing [35], ASHA workers are not currently trained to perform testing themselves [36]. However, at least 3 large community-based studies in India have shown the strong feasibility of training ASHA-level workers to conduct capillary blood glucose measurements and the trust that community members have in ASHA-administered testing [37–39]. Therefore, it is possible that having ASHA workers perform the OGTT themselves may streamline GDM testing and counseling and

reduce the number of clinic visits. Our results suggest that integrating GDM screening into ASHA workflow could be both feasible and beneficial. Further studies are needed to determine whether such an approach would be cost-effective and scalable.

Our study also identified a surprisingly high prevalence of GDM in these low-income populations, higher than the previously reported Indian national average [31, 40]. Previous studies of GDM in India have identified traditional risk factors such as age, BMI and belonging to a higher socioeconomic status [40]. However, our study population did not have these risk factors. The median age was only 24 years, whereas >35 years is generally considered high risk for GDM. Furthermore, the median BMI in our study was <25 kg/m$^2$ while a BMI of >25 kg/m$^2$ is traditionally considered to be a risk factor for GDM [41]. This means that most women in this study would have been missed by age and BMI criterion. Our qualitative data suggest that the high prevalence of GDM in slum communities is also unknown to local health care workers, who perceived that low-income populations are not at risk for GDM. Clinicians stated that they screen women for GDM based on their risk profile, and only offer testing to women accordingly. Therefore, most women in this study would have been missed by the traditional risk profile. These findings highlight the importance of providing accessible standardized interventions to low-income populations, who are at equal or higher risk that higher income populations, but may not be considered so in current clinical practice.

In our study, linkage to clinical care after a GDM diagnosis was nearly 100% (30 out of 31 women). However, management of women with GDM varied significantly based on clinic and provider. Current national and international guidelines recommend lifestyle and dietary counseling for all women who are diagnosed with GDM. Yet more than 60% of women diagnosed with GDM who visited a clinic did not receive counseling or medication. This unfortunate gap in the transition from screening to clinical care has also been seen in CHW-based interventions for other diseases. In a study in Tanzania, CHWs identified women with HIV who would benefit from early antenatal care, but there was no significant increase in the actual attendance of antenatal clinic visits [42]. Similarly, a CHW-based hypertension intervention in Kenya improved linkage to care but did not find a statistically significant improvement in blood pressure control, possibly because of the health system's focus on acute rather than chronic disease treatment [43, 44]. In our study, adequate counseling may have been hindered by clinicians' perceived lack of time to provide counseling and a misunderstanding of who "high-risk" populations are in their community. Future interventions should focus on addressing these barriers at the health system-level via clinician education to strengthen appropriate management of care following linkage to clinical care. Clinician interviews also reflected another challenge of clinic-based care in India: women often attend clinic visits without family, but disease counseling is more effective when family is involved [45]. Through home-based testing, CHWs–who have equal or greater effectiveness in counseling compared to clinicians [46]–have the ability to counsel the individual and the household, which not only offloads the burden of counseling from clinics but also could improve adherence and sustainability of linkage to care.

Our study had several strengths and limitations. We conducted a well-powered study to estimate the uptake of a CHW-delivered GDM screening intervention and documenting the reasons for non-uptake. Collecting more data on participants who declined the intervention may have improved our understanding of the barriers to uptake of screening, though there were very few who declined. For this study, we screened for GDM using capillary blood glucose measurements. Capillary blood may lead to overdiagnosis but overall correlates well with venous blood glucose (r = 0.80) [47]. Therefore, the benefit of convenience and feasibility of capillary testing in a high-risk population outweighs the risk of overdiagnosis in women who would otherwise not have been screened at all. Furthermore, confirmatory testing using

venous blood samples should be done in clinical settings. We included multiple stakeholders —pregnant women, CHWs, and clinicians from public and private hospitals in obstetrics and medicine—in our qualitative analysis, allowing for a broad understanding of stakeholder perspectives. We selected clinicians from both public and private clinics and hospitals to mitigate selection bias and repeated questions throughout the interview to mitigate bias in information recall. We sampled two large urban slum communities, though these data may be limited in applicability to more rural slums. Through this approach, we found that CHW-delivered services could include a broad range of preventive health services [29], within which GDM screening should be integrated.

## Conclusion

CHWs were able to deliver gold standard GDM screening to 90% of low-income pregnant women living in two urban slums in India. CHW-delivered testing overcame logistical barriers to screening for pregnant women and CHWs were found to have unique social capital that allowed improvement of community health beyond their logistical impact. However, linkage to appropriate clinical management may have been compromised by a misunderstanding by health care workers of the risk factors for GDM in slum communities. Increasing awareness of GDM in slum communities through CHW-delivered GDM screening and counseling may serve as a model for decentralizing preventive health care during pregnancy, which would off-load busy public clinics and expand access to care for marginalized populations.

## Supporting information

**S1 Table. Characteristics of women who underwent OGTT.**
(DOCX)

**S1 Text. Interview guide for participants.**
(DOCX)

**S2 Text. Interview guide for CHWs.**
(DOCX)

## Author Contributions

**Conceptualization:** Puja Chebrolu, Andrea Chalem, Matthew Ponticiello, Jyoti S. Mathad, Radhika Sundararajan.

**Data curation:** Andrea Chalem, Matthew Ponticiello.

**Formal analysis:** Puja Chebrolu, Andrea Chalem, Matthew Ponticiello, Kathryn Broderick, Arthi Vaidyanathan, Rachel Lorenc, Jyoti S. Mathad, Radhika Sundararajan.

**Funding acquisition:** Puja Chebrolu, Andrea Chalem, Matthew Ponticiello, Jyoti S. Mathad, Radhika Sundararajan.

**Methodology:** Puja Chebrolu, Matthew Ponticiello, Vaishali Kulkarni, Ashlesha Onawale, Jyoti S. Mathad, Radhika Sundararajan.

**Supervision:** Puja Chebrolu, Andrea Chalem, Matthew Ponticiello, Vaishali Kulkarni, Ashlesha Onawale, Jyoti S. Mathad, Radhika Sundararajan.

**Writing – original draft:** Puja Chebrolu, Andrea Chalem, Matthew Ponticiello, Kathryn Broderick, Arthi Vaidyanathan, Radhika Sundararajan.

**Writing – review & editing:** Puja Chebrolu, Andrea Chalem, Matthew Ponticiello, Kathryn Broderick, Arthi Vaidyanathan, Rachel Lorenc, Vaishali Kulkarni, Ashlesha Onawale, Jyoti S. Mathad, Radhika Sundararajan.

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
