## [Decision Letter · Decision Letter 0]

6 Mar 2023

PGPH-D-22-02116

A community health worker-led program to improve access to gestational diabetes screening in urban slums of Pune, India: results from a mixed methods study

Dear Dr. Chebrolu,

Thank you for submitting your manuscript to PLOS Global Public Health. Firstly, we would like to apologize for the delay in processing your manuscript. It has been exceptionally difficult to secure reviewers to evaluate your study. We have now received one completed review, which is available below. The reviewer has raised significant scientific concerns about the study that need to be addressed in a revision.

Please note that we have only been able to secure a single reviewer to assess your manuscript. We are issuing a decision on your manuscript at this point to prevent further delays in the evaluation of your manuscript. Please be aware that the editor who handles your revised manuscript might find it necessary to invite additional reviewers to assess this work once the revised manuscript is submitted. However, we will aim to proceed on the basis of this single review if possible. 

After careful consideration, we feel that it has merit but does not fully meet PLOS ONE’s publication criteria as it currently stands. Therefore, we invite you to submit a revised version of the manuscript that addresses the points raised during the review process.

We look forward to receiving your revised manuscript.

Kind regards,

Miquel Vall-llosera Camps

Staff Editor

Journal Requirements:

1. Please send a completed 'Competing Interests' statement, including any COIs declared by your co-authors. If you have no competing interests to declare, please state "The authors have declared that no competing interests exist". Otherwise please declare all competing interests beginning with the statement "I have read the journal's policy and the authors of this manuscript have the following competing interests:"

a. State what role the funders took in the study. If the funders had no role in your study, please state: “The funders had no role in study design, data collection and analysis, decision to publish, or preparation of the manuscript.”

b. If any authors received a salary from any of your funders, please state which authors and which funders.

3. We do not publish any copyright or trademark symbols that usually accompany proprietary names, eg  ©, ®, ™  (e.g. next to drug or reagent names). Please remove all instances of trademark/copyright symbols throughout the text, including ® on page 4.

4. Fig 1: please (a) provide a direct link to the base layer of the map (i.e., the country or region border shape) and ensure this is also included in the figure legend; and (b) provide a link to the terms of use / license information for the base layer image or shapefile. We cannot publish proprietary or copyrighted maps (e.g. Google Maps, Mapquest) and the terms of use for your map base layer must be compatible with our CC-BY 4.0 license. 

Additional Editor Comments:

We would expect qualitative and mi xed-methods studies to include the following: 1) defined objectives or research questions; 2) description of the sampling strategy, including rationale for the recruitment method, participant inclusion/exclusion criteria and the number of participants recruited; 3) detailed reporting of the data collection procedures; 4) data analysis procedures described in sufficient detail to enable replication; 5) a discussion of potential sources of bias; and 6) a discussion of limitations.

Reviewers' comments:

Reviewer's Responses to Questions

**Comments to the Author**

1. Does this manuscript meet PLOS Global Public Health’s publication criteria? Is the manuscript technically sound, and do the data support the conclusions? The manuscript must describe methodologically and ethically rigorous research with conclusions that are appropriately drawn based on the data presented.

Reviewer #1: Yes

2. Has the statistical analysis been performed appropriately and rigorously?

Reviewer #1: N/A

3. Have the authors made all data underlying the findings in their manuscript fully available (please refer to the Data Availability Statement at the start of the manuscript PDF file)?

Reviewer #1: No

4. Is the manuscript presented in an intelligible fashion and written in standard English?

Reviewer #1: Yes

5. Review Comments to the Author

Reviewer #1: the attached file contains hyperlinks for the authors convenience which I don't think are contained in the comments below

The authors describe a mixed methods study of a CHW led diabetes screening program to enable detection of GDM in pregnant women in urban slum area.

Overall, the study reveals some interesting results and insights which will inform improved detection of screening for GDM in women in India.

I have specific comments below which I believe will clarify aspects of their study for the reader.

I’m not sure that WHO does in fact recommend all women for screening for GDM by OGTT, they certainly suggest maternal assessment for hyperglycaemia, however in your ref #4 the suggestion is that GDM can be diagnosed with a FPG > 5.1 mmol/L or the two hr OGTT, before 24 – 28 weeks (so late 2nd early 3rd trimester) so I am suggesting this be toned down a bit to truly reflect the WHO guidelines noting as well that WHO do in fact recommend task shifting health promotion, nutritional supplements and IPTp, which you could argue your study fulfills these recommendations by WHO for task shifting see anc-guideline-presentationb8005106-0947-4329-adb1-61aea46db1db.pptx (live.com)

The description of sampling and recruitment requires a bit more detail, how big were the study areas, did they go door to door to every single dwelling in the study areas or was there a more household sampling approach, what was the denominator population (ie women of CBA, or some way to estimate reach of the CHW recruitment)

How were the CHW trained to collect anthropometry and BP as these are skills that need to be acquired after appropriate training – seems like they only received training for the OGTT, what were there skill levels for other clinical assessments – were they ASHA, or angawadi or some other specific cadre. How were they trained to deliver counselling about healthy diet etc where did this health promotion information come from or was it developed for the study

How was gestational age assessed during the recruitment process to determine the woman was more than 24 weeks pregnant.

How was BMI calculated during the pregnancy? It is not as simple as for non pregnant women (wgt/height2)

How far away were “the nearby antenatal clinic” to which the women were referred. As there is discussion about logistics to get to antenatal care it’s reasonable to want to know how far women might need to travel to access this care (time or distance).

Where was the REDCap server and dta held, who was the custodian or the data was that the Deep Griha Society, What was the role of this society in the collaboration.

What measures were taken to ensure the safety of the water in which the OGTT was prepared?, was this just regular household drinking water?

What was the actual referral process? Were they just advised to go to the clinic or were they given appointments or letters of referral for the health centre/clinic.

Why was ethics not necessary or obtained from ICMR particularly if it was an overseas based study, having conducted a similar task shifting study in India we were required to obtained ethical approval from ICMR.

There are no sample/power calculations described, although you mention in the discussion of strengths that it was a well powered study?

Results

Reference required for the SES scale used (Kuppuswamy socioeconomic status)

13% of women had hypertension, what was done for these women, were any of these women also GDM?

You state reasons for “eligible” women not completing OGTT was previous OGTT (Figure 3), and yet in the eligibility criteria these women should have been excluded?

You mention several times the “logistics” as a barrier to attending the clinic, but I only read mention of travel, would be good to outline this a bit more, particularly in the context of time and distance (so a bit more explanation about how far the women were from health centres/how much transport might cost)

I find the quote on line 225 a bit strange, had she had OGTT before how did she know it took all day, how was she aware of the OGTT before the CHW came, did she have a history of GDM?

Figure 2 study flow chart – you need to describe why the 10% of eligible women were not enrolled in the figure,

You detected 14% of women with GDM and describe it as “surprisingly high prevalence” – however in the background (reference #1) you state up to 40% and then in the discussion state this 14% was higher than the previously reported national average (reference #27). These two sentences and references appear inconsistent.

Line 330, the discussion of BMI is a bit confusing whilst the BMI of the entire cohort was 23.8, the median BMI of women with GDM was more than 25.5 which confirms the risk profile. BMI is not the most reliable assessment of risk though.

In general the quality of the figures is low, can you provide better quality images?

Figure 1.

Is it possible perhaps to include health centres in this figure and provide a bit more detail

Figure 2 provide reasons for unable to be enrolled, how many just refused without reason/not interested.

Figure 3 as previously mentioned the 34.8% with a previous OGTT should have been excluded shouldn’t they as they did fit the inclusion criteria

Figure 4 – poor quality difficult to read even when zoomed in numbers are all blurry

Supplemental table 1 would be better in the actual manuscript deleting Table 1 currently in main manuscript.

You could provide the Interview questions only for the participants/CHW/ and Clinicians, you do not have to provide any responses but it is important to know what questions were asked during the interviews

References

#4 the link provided does not direct the reader to the full article rather a summary. Link should be changed to Diagnostic criteria and classification of hyperglycaemia first detected in pregnancy: A World Health Organization Guideline (diabetesresearchclinicalpractice.com).

Also if you wish to reference the summary why not use the updated version (august 2021)

WHO recommendation on the diagnosis of gestational diabetes in pregnancy (srhr.org)

6. PLOS authors have the option to publish the peer review history of their article (what does this mean?). If published, this will include your full peer review and any attached files.

**Do you want your identity to be public for this peer review?** For information about this choice, including consent withdrawal, please see our Privacy Policy.

Reviewer #1: No

---

## [Decision Letter · Decision Letter 1]

18 Jun 2023

PGPH-D-22-02116R1

A community health worker-led program to improve access to gestational diabetes screening in urban slums of Pune, India: results from a mixed methods study

Dear Dr. Chebrolu,

Thank you for submitting your manuscript to PLOS Global Public Health. After careful consideration, we feel that it has merit but does not fully meet PLOS Global Public Health’s publication criteria as it currently stands. Therefore, we invite you to submit a revised version of the manuscript that addresses the points raised during the review process.

Please see the comments from two reviewers below. A new reviewer was invited at R1, and they've tried to not go much beyond the previous round of review, but have flagged a number of concerns that they feel will help increase the validity and impact of the study.

We look forward to receiving your revised manuscript.

Kind regards,

Hanna Landenmark

Staff Editor

Journal Requirements:

Additional Editor Comments (if provided):

Reviewers' comments:

Reviewer's Responses to Questions

**Comments to the Author**

1. If the authors have adequately addressed your comments raised in a previous round of review and you feel that this manuscript is now acceptable for publication, you may indicate that here to bypass the “Comments to the Author” section, enter your conflict of interest statement in the “Confidential to Editor” section, and submit your "Accept" recommendation.

Reviewer #1: All comments have been addressed

Reviewer #2: (No Response)

2. Does this manuscript meet PLOS Global Public Health’s publication criteria? Is the manuscript technically sound, and do the data support the conclusions? The manuscript must describe methodologically and ethically rigorous research with conclusions that are appropriately drawn based on the data presented.

Reviewer #1: Yes

Reviewer #2: Partly

3. Has the statistical analysis been performed appropriately and rigorously?

Reviewer #1: N/A

Reviewer #2: Yes

4. Have the authors made all data underlying the findings in their manuscript fully available (please refer to the Data Availability Statement at the start of the manuscript PDF file)?

Reviewer #1: No

Reviewer #2: No

5. Is the manuscript presented in an intelligible fashion and written in standard English?

Reviewer #1: Yes

Reviewer #2: Yes

6. Review Comments to the Author

Reviewer #1: all comments addressed adequately thank you

Reviewer #2: The authors, in my assessment, have tried their best to address the concerns from the previous round of review. Unfortunately, I was not part of that review team. So, my questions below may appear to be new to the authors. However, I do think that they are important and should be addressed (at least in writing) before the paper can be accepted for publication.

While I see the benefits of task-shifting and several studies in the past have shown the benefits of this approach across a spectrum of public health contexts and clinical conditions, I am afraid that there is a limit to how much work can be shifted to lay health workers and how that work will be integrated and coordinated.

In India, we are already have a cadre of community health workers (ASHAs) who are tasked with many activities within maternal and child health domain. One of these tasks include motivating the expectant mother to attend antenatal visit and accompany her for the visit to the health facility. In fact, MoHFW guidelines clearly highlight the goal of universal testing for GDM among all pregnant women and lays out the operational guidelines to achieve this goal including the responsibilities of ASHA workers. See here: https://nhm.gov.in/New_Updates_2018/NHM_Components/RMNCH_MH_Guidelines/Gestational-Diabetes-Mellitus.pdf

Could the authors comment on how their approach is either consistent with the guidelines or whether their findings imply modification of the guidelines? If the latter, what would be the cost-effectiveness of this approach and how would this approach fit in within the overall workflow of ASHAs with regards to caring for pregnant women? Which of the tasks (other than GDM screening) should be shifted to the community from the health facility? My concern is that, studied in isolation, each task may be better off shifted to the community, but that it may be infeasible to do so when looked at in totality.

7. PLOS authors have the option to publish the peer review history of their article (what does this mean?). If published, this will include your full peer review and any attached files.

**Do you want your identity to be public for this peer review?** For information about this choice, including consent withdrawal, please see our Privacy Policy.

Reviewer #1: No

Reviewer #2: No

---

## [Editor Report · Decision Letter 2]

23 Aug 2023

PGPH-D-22-02116R2

A community health worker-led program to improve access to gestational diabetes screening in urban slums of Pune, India: results from a mixed methods study

Dear Dr. Chebrolu,

Thank you for submitting your manuscript to PLOS Global Public Health. After careful consideration, we feel that it has merit but does not fully meet PLOS Global Public Health’s publication criteria as it currently stands. Therefore, we invite you to submit a revised version of the manuscript that addresses the points raised during the review process.

We look forward to receiving your revised manuscript.

Kind regards,

Sarang Deo, PhD

Academic Editor

Journal Requirements:

Additional Editor Comments (if provided):

I see that the authors have attempted to respond to the comments from the new reviewer in the last round. The response and associated edits in the manuscript are mostly satisfactory. However, I would like the authors to add a little more detail on ASHA's ability to conduct OGTT by themselves based on (i) their training and skill level, (ii) government regulations, (iii) willingness of the patient to undergo the test with ASHA. I suggest that the authors look for references in the literature, where similar interventions done by ASHAs are evaluated. I would also like the authors to be less definitive about the time saved of ASHAs if they do the OGTT themselves and explicitly leave it for a future research study to establish this rigorously. I think the paper should be ready for acceptance once these are incorporated in the new draft.
---

## [Editor Report · Decision Letter 3]

3 Oct 2023

A community health worker-led program to improve access to gestational diabetes screening in urban slums of Pune, India: results from a mixed methods study

PGPH-D-22-02116R3

Dear Dr. Chebrolu,

We are pleased to inform you that your manuscript 'A community health worker-led program to improve access to gestational diabetes screening in urban slums of Pune, India: results from a mixed methods study' has been provisionally accepted for publication in PLOS Global Public Health.

Best regards,

Sarang Deo, PhD

Academic Editor

The authors have addressed the residual comments. I don’t have any additional requests and recommend that the paper be accepted for publication.